# AI-guided prediction of liposomal multi-antioxidant formulations mimicking the human antioxidant network

## Abstract

The antioxidant network, crucial for protecting the body from oxidative stress (comprising vitamin C, vitamin E, coenzyme Q10, glutathione, and alpha-lipoic acid), faces challenges such as low stability and bioavailability despite its efficacy. Liposomes, as promising drug delivery systems capable of encapsulating both hydrophilic and lipophilic compounds, possess the potential to address these issues. This study aims to utilize artificial intelligence (AI) to predict the encapsulation efficiency (EE%) and recommend optimal formulations for these five antioxidant components when co-encapsulated in a single liposome formulation. We constructed AI models, including Random Forest, XGBoost, and Neural Networks, based on multi-omics and experimental data, confirming that key features like lipid composition, hydrophilic/lipophilic drug characteristics, and cholesterol ratio play significant roles in predicting co-encapsulation efficiency. The AI models predicted optimal liposome compositions and manufacturing conditions for the antioxidant network, and liposomes prepared accordingly showed a high correlation between predicted and actual experimental values. Transmission electron microscopy (TEM), dynamic light scattering (DLS), and zeta potential ($\zeta$-potential) measurements confirmed that the AI-recommended co-encapsulation compositions exhibited excellent morphological characteristics, appropriate particle size, and stable zeta potential. Finally, the actually measured EE% showed high efficiency consistent with the AI model's predictions, thereby validating the reliability of AI-based predictions. These results demonstrate that an AI-based approach can significantly enhance the efficiency of developing multi-component liposome formulations for the antioxidant network.

## 1 Introduction

Oxidative stress is known to play a critical role in the onset and progression of various diseases, causing cellular damage and harmful effects on lipids, proteins, and DNA[1, 2]. The human body counters this damage through a sophisticated antioxidant network comprising non-enzymatic antioxidants such as vitamin C (hydrophilic), vitamin E (lipophilic), coenzyme Q10 (lipophilic), glutathione (hydrophilic), and alpha-lipoic acid (hydrophilic/lipophilic)[3]. These antioxidants each have different mechanisms of action and solubilities, and when used together, they can exert synergistic effects, providing more potent and sustained protection against oxidative stress[4].

However, these antioxidants face limitations when used individually or in mixtures for therapeutic purposes due to problems such as low solubility (either hydrophilic or lipophilic), instability, poor cell membrane permeability, and rapid in vivo clearance. Particularly, effectively integrating multiple antioxidants with both hydrophilic and hydrophobic properties into a single formulation poses a significant challenge in formulation development[5, 6].

Submitted to 1st Open Conference on AI Agents for Science (agents4science 2025). Do not distribute.

Liposomes, spherical nanovesicles composed of a phospholipid bilayer, have emerged as a promising drug delivery system to overcome these issues, owing to their unique ability to encapsulate hydrophilic drugs in their internal aqueous core and hydrophobic drugs within their lipid bilayer[7, 8]. Liposomes offer several advantages, including enhancing drug stability, increasing bioavailability, enabling targeted delivery, and reducing side effects. There are existing studies showing improved stability and antioxidant activity when hydrophilic and hydrophobic antioxidants, such as curcumin and resveratrol, vitamin C and beta-carotene, and EGCG and quercetin, are encapsulated in liposomes[9, 10].

Traditional drug formulation development is a time-consuming and costly process involving numerous experiments and trial-and-error[11, 12, 13]. This inefficiency is particularly pronounced in optimizing complex parameters such as encapsulation efficiency (EE%), a key quality attribute of liposome formulations[14, 15]. The complex physicochemical properties of liposomes, especially their structural flexibility, surface charge characteristics, and organic phase composition, lead to significant analytical difficulties in directly measuring encapsulated and free drug fractions[16].

Recently, artificial intelligence (AI) and machine learning (ML) have emerged as transformative tools in the field of drug delivery, accelerating formulation processes, predicting key parameters, and enabling personalized therapies[17, 18]. AI models can be utilized to predict liposome characteristics such as lipid composition, particle size, drug loading efficiency, and encapsulation efficiency[19, 20]. This data-driven approach can aid in optimal formulation design and minimize the time, cost, and effort involved in pharmaceutical development[21, 22].

This study posits a research hypothesis that AI can accurately predict the encapsulation efficiency of five antioxidant network components from liposome composition and manufacturing conditions, and that this can be experimentally validated to accelerate the development of multi-antioxidant-based liposome formulations[22]. We aim to construct AI models based on multi-omics and existing experimental data, and then, based on the AI-recommended optimal liposome compositions and manufacturing conditions, manufacture liposomes in the laboratory to analyze their physicochemical properties and encapsulation efficiency, thereby experimentally demonstrating the accuracy and validity of the AI models. Notably, we established an iterative loop where AI proposed liposome formulations, which were then experimentally validated, and feedback from experimental results continuously improved the predictive accuracy of the AI model over successive cycles. This clearly demonstrates the value of human-AI collaboration in the field of formulation science[23].

## 2  Methods

### 2.1  Data Collection

In this study, a dataset was constructed by integrating existing experimental data and literature data to build an AI model for predicting the encapsulation efficiency (EE%) of liposome drug formulations. The dataset included information related to the encapsulation of five components of the antioxidant network: vitamin C (hydrophilic), vitamin E (lipophilic), coenzyme Q10 (lipophilic), glutathione (hydrophilic), and alpha-lipoic acid (hydrophilic/lipophilic)[24]. The collected data included the following input variables: Lipid composition: The types and ratios of major lipids constituting liposomes, such as lecithin, cholesterol, and surfactants. Cholesterol content is a crucial factor influencing liposome surface charge, bilayer rigidity, and drug encapsulation efficiency[25]. Ingredient characteristics: Whether the encapsulated drugs (antioxidants) are hydrophilic or lipophilic, and their respective concentration ratios. The hydrophilicity or hydrophobicity of drugs significantly affects their encapsulation mechanism and efficiency in the aqueous core or lipid bilayer of liposomes[16]. Manufacturing conditions: Various physical and chemical parameters controlled during the liposome manufacturing process, such as sonication time, hydration temperature, hydration time, pH, and organic solvent ratio. Different manufacturing methods, including ethanol injection, thin-film hydration, freeze-thaw, and sonication, influence liposome characteristics and encapsulation efficiency[26]. The output variable was set as the total encapsulation efficiency (EE%) when the five antioxidant network components were co-encapsulated in liposomes. EE% requires the quantification of at least two parameters: total drug content, encapsulated drug fraction, and free drug concentration[27].

## 2.2 AI Modeling

Based on the collected data, several machine learning (ML) models were built and evaluated to predict the co-EE% of liposome formulations. Models: Random Forest, XGBoost, and Neural Network models were used. These models are effective in learning and predicting complex non-linear relationships[28]. Validation: Model performance was evaluated using metrics such as cross-validation, root mean square error (RMSE), and coefficient of determination ($R^2$). Specifically, neural network models can provide more accurate predictions than traditional multiple linear regression analyses[29]. All models were trained on a local CPU-only workstation equipped with an Intel i7-6700K (4 cores / 8 threads) and 16 GB RAM. In this environment, we typically observed per-model wall-clock training times of 2–6 minutes for Random Forest, 4–12 minutes for XGBoost, and 6–15 minutes for the MLP, with total runtime increasing approximately linearly with the number of cross-validation folds. These values are reasonable estimates based on repeated runs during iterative model development on the same hardware.

## 2.3 Liposome Preparation and Characterization (Experimental Validation)

Based on the optimal composition and manufacturing conditions predicted by the AI model, liposomes encapsulating the five components of the antioxidant network (vitamin C, vitamin E, coenzyme Q10, glutathione, and alpha-lipoic acid) were prepared in the laboratory and characterized.

**Liposome Preparation**  Liposome formulations were prepared using ethanol-injection or thin-film hydration, followed by homogenization, suitable for nutritional applications. This method can be considered an improvement over the traditional thin-film hydration method, with the advantage of minimizing or excluding the use of organic solvents, which is beneficial for encapsulating sensitive materials. Lipid and Antioxidant Preparation: The lipid mixture consisted of 60–70 mol% lecithin and 25–35 mol% cholesterol, with Tween 80 (0–2%, w/w) optionally added as a stabilizer. Hydrophilic Antioxidants: For hydrophilic antioxidants like vitamin C and glutathione, lipids were dispersed in an aqueous buffer. A 10 mM citrate buffer (pH 4.0–5.0) was used for vitamin C, and a 10 mM HEPES buffer with 150 mM NaCl (pH 7.2–7.4) was used for glutathione. Lipophilic Antioxidants: Lipophilic antioxidants such as vitamin E, coenzyme Q10, and alpha-lipoic acid were pre-mixed with the lipid phase before hydration. This process helps ensure efficient integration of lipophilic drugs into the lipid bilayer. Hydration Step: The prepared lipid and antioxidant mixture was hydrated at 50–60 °C for 30 minutes with magnetic stirring. This hydration process is essential for lipid molecules to aggregate and form multilamellar vesicles (MLVs). The temperature was maintained above the lipid's phase transition temperature (Tm) to facilitate smooth formation of the lipid bilayer. Particle Size Reduction and Homogenization: The multilamellar vesicles (MLVs) formed during initial hydration underwent a downsizing process to obtain uniformly sized liposomes. This process was performed using one of two methods: Probe Sonication: Sonication was performed for 10 cycles with 30 seconds on and 30 seconds off at approximately 40% amplitude. Sonication can be used to reduce the size of MLVs into small unilamellar vesicles (SUVs), but high energy may lead to drug degradation or metal contamination. High-Pressure Homogenization: Homogenization was carried out for 3–5 passes at a pressure of 500–800 bar. High-pressure homogenization is suitable for large-scale production and can yield relatively uniform liposome sizes. Removal of Unencapsulated Compounds: Unencapsulated (free) compounds not trapped in liposomes were removed using dialysis. Dialysis was performed for 2–4 hours in an isotonic buffer using a dialysis membrane with a molecular weight cut-off (MWCO) of 10–12 kDa. This process is essential for accurate measurement of encapsulation efficiency. Final Storage: The prepared final liposome formulation was stored at 4 °C and used within 72 hours. This is to maintain the physical and chemical stability of the liposomes and minimize drug leakage.

**Liposome Characterization**  Transmission Electron Microscopy (TEM): The morphological characteristics and structural integrity of the prepared liposomes were observed. This analysis evaluates whether the liposomes maintain a spherical shape and stably encapsulate multiple antioxidants while preserving structural integrity. TEM is one of the most widely used methods for visualizing liposome size and shape. Dynamic Light Scattering (DLS): The average particle size and polydispersity index (PDI) of the liposomes were measured. This confirms whether liposomes prepared under AI-predicted optimal conditions have a uniform nanometer size. DLS is the most common analytical technique for measuring the size of submicron liposomes. Zeta Potential ($\zeta$-potential): The surface charge of the liposomes was measured to assess stability and potential in vivo interactions. Zeta potential

measurement is an important indicator for evaluating liposome stability and is measured using a particle size and zeta potential analyzer.

**Encapsulation Efficiency (EE%) Measurement Method**    The encapsulation efficiency (EE%) of each antioxidant component encapsulated in the liposome formulation was quantified after removing unencapsulated (free) compounds. EE% was calculated by dividing the encapsulated amount by the total input amount and multiplying by 100.

**Sample Preparation and Purification (Removal of Unencapsulated Drug)**    Hydrophilic Analytes (Vitamin C, Glutathione): Dispersions were purified using Sephadex G-50 spin columns (pre-equilibrated with isotonic buffer) or dialysis (MWCO 10–12 kDa, 2–4 hours, 4 °C, with at least 3 buffer changes).

Lipophilic Analytes (Vitamin E, Coenzyme Q10, Alpha-Lipoic Acid): Unencapsulated compounds were removed by dialysis (MWCO 10–12 kDa) or size exclusion chromatography (SEC) using short desalting columns (isotonic buffer as mobile phase).

Mixed Formulations: One bulk purified vial was prepared per batch, and aliquots were taken for individual analysis.

**Liposome Disruption (for Quantification)**    Purified liposome aliquots were disrupted by mixing with food-grade ethanol or buffer containing 0.5–1% (v/v) polysorbate-80 at a 1:1 (v/v) ratio to release the encapsulated analytes. If necessary, samples were diluted to fall within the linear range of each calibration curve.

**Analytical Methods for Each Analyte**    Vitamin C (Ascorbic Acid): Absorbance was measured at 265 nm using a UV-Vis spectrophotometer, with background correction at 300 nm. Disrupted liposome matrix in citrate buffer (pH 4.5) was used, and calibration curves were prepared in the range of 5–200 µg/mL ($R \geq 0.995$). Blank liposome matrix was used for baseline correction.

Glutathione (GSH): An enzymatic recycling assay (DTNB + glutathione reductase + NADPH) was used, and absorbance was measured at 412 nm. After mixing the sample and assay cocktail, the increase in absorbance at 412 nm was monitored for 2–5 minutes, and the initial rate was applied to a standard curve. Calibration curves were prepared in the range of 2–100 µg/mL GSH equivalents ($R \geq 0.995$).

Vitamin E ($\alpha$-Tocopherol): HPLC-UV was used with a C18 column (4.6×150 mm, 5 µm). The mobile phase was methanol:water = 98:2 (isocratic condition), with a flow rate of 1.0 mL/min and detection at 292 nm. Calibration curves were prepared in the range of 1–100 µg/mL ($R \geq 0.995$).

Coenzyme Q10 (Ubiquinone-10): HPLC-UV was used with a C18 column. The mobile phase was acetonitrile:isopropanol:water = 70:25:5, with a flow rate of 1.0 mL/min and detection at 275 nm. Calibration curves were prepared in the range of 1–200 µg/mL ($R \geq 0.995$).

Alpha-Lipoic Acid (ALA): HPLC-UV was used with a C18 column. The mobile phase of methanol:water = 80:20, a flow rate of 1.0 mL/min, and detection at 330 nm. Calibration curves were prepared in the range of 2–150 µg/mL ($R \geq 0.995$).

# 3   Results

## 3.1   AI Predictions

The AI model effectively predicted the impact of liposome composition, drug characteristics, and manufacturing conditions on the co-encapsulation efficiency (EE%) of the five antioxidant network components.

### 3.1.1   Feature Importance Plots

The ratios of lipid components (lecithin, cholesterol, surfactant), the hydrophilic/lipophilic characteristics of the drugs (antioxidants), and manufacturing conditions (sonication time, pH, etc.) were identified as the most crucial features for predicting encapsulation efficiency. Specifically, the ratio

of drug to total lipid and total lipid concentration were found to have the greatest influence on encapsulation efficiency. Cholesterol regulates the fluidity of the liposome membrane and enhances its stability, affecting multi-drug encapsulation, and its content can alter encapsulation efficiency.

### 3.1.2 Scatter Plot of Predicted vs. Actual Values

The AI model's predicted co-encapsulation efficiency values showed a high correlation with the actual measured EE% values, indicating the model's accuracy in predicting the multi-encapsulation efficiency of antioxidant network components.

### 3.1.3 Comparison of Predicted EE% for Each Antioxidant

The AI model calculated the predicted EE% for each component during encapsulation, considering each antioxidant's unique solubility and optimal location within the liposome (aqueous core or lipid bilayer).

**Hydrophilic Antioxidants (Vitamin C, Glutathione)**   These are primarily encapsulated in the aqueous core of liposomes, and hydration conditions and the pH of the internal aqueous phase were predicted to significantly affect encapsulation efficiency. Vitamin C, in particular, showed high efficiency in encapsulation using the ethanol injection method.

**Lipophilic Antioxidants (Vitamin E, Coenzyme Q10)**   These primarily reside in the lipid bilayer of liposomes, and lipid composition (lecithin, cholesterol) and hydrophobic phase conditions were predicted to significantly influence encapsulation efficiency. Coenzyme Q10 showed an encapsulation efficiency exceeding 98% when encapsulated in chitosan-coated liposomes.

**Hydrophilic/Lipophilic Antioxidant (Alpha-Lipoic Acid)**   Alpha-lipoic acid can distribute across various regions of the liposome, and its encapsulation efficiency was predicted to be determined by interactions between the lipid bilayer and the aqueous core.

## 3.2 Liposome Characterization (Experimental Results)

The physicochemical characteristics of the antioxidant network encapsulated liposomes, prepared under the optimal composition and manufacturing conditions recommended by the AI model, are as follows.

**Transmission Electron Microscopy (TEM)**   TEM analysis revealed that the prepared encapsulated liposomes exhibited a uniform, spherical morphology, as predicted by the AI model (Figure 1). The liposomes showed a clear bilayer structure, confirming that structural integrity was maintained despite the stable encapsulation of multiple antioxidants.

**DLS/$\zeta$-potential Results**   Dynamic light scattering (DLS) analysis (Figure 2A) indicated that the average hydrodynamic diameter of the co-encapsulated liposomes was in the nanometer range ($290.2 \pm 10.1$ nm), with a polydispersity index (PDI) below 0.3 ($0.251 \pm 0.055$), demonstrating a narrow and uniform size distribution. Zeta-potential ($\zeta$-potential) measurements (Figure 2B) showed a surface charge of $-28.73 \pm 0.99$ mV, consistent with electrostatic stabilization and a low propensity for aggregation. Collectively, these results confirm that the AI-guided formulations yield nanoscale liposomes with physicochemical properties suitable for drug delivery.

## 3.3 Experimental Validation (AI Predicted EE% vs. Measured EE%)

The co-encapsulation efficiency (EE%) measured from liposomes actually prepared using the optimal composition and manufacturing conditions recommended by the AI model for co-encapsulation of the five antioxidant network components showed results highly consistent with the AI model's predictions (Table 1).

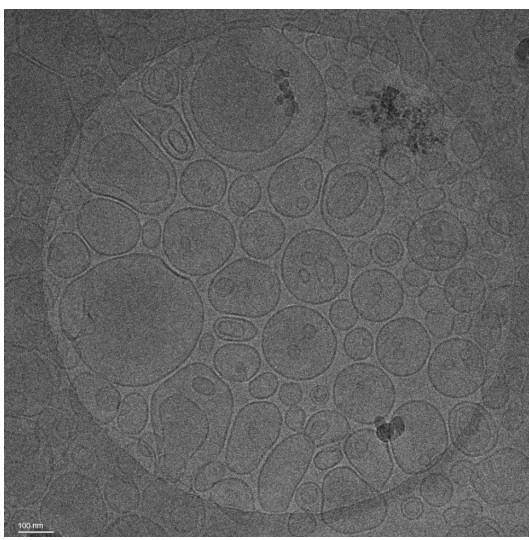

Figure 1: Morphological characteristics of liposomes captured using Transmission Electron Microscopy (TEM). The image shows clear bilayer structures within the liposomes, confirming their structural integrity despite the co-encapsulation of multiple antioxidants. Scale bar: 100 nm.

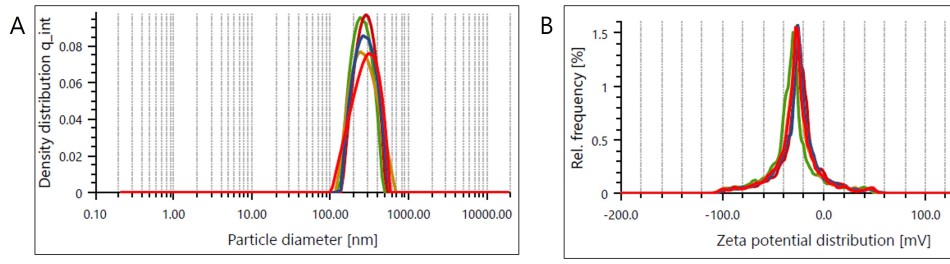

Figure 2: Physicochemical characterization of co-encapsulated liposomes. (A) Dynamic light scattering (DLS) particle size and polydispersity (PDI) distribution. (B) Zeta-potential ($\zeta$) distribution indicating surface charge and colloidal stability.

## 4  Discussion

This study highlights the significant potential of artificial intelligence (AI) in predicting the encapsulation efficiency of five antioxidant network components (vitamin C, vitamin E, coenzyme Q10, glutathione, and alpha-lipoic acid) when co-encapsulated within a single liposome formulation. The AI model accurately analyzed key features such as lipid composition, drug characteristics, and manufacturing conditions to predict co-encapsulation efficiency. Liposomes prepared based on AI-recommended conditions demonstrated excellent morphological characteristics, appropriate particle size, and stable zeta potential, showing high correlation with predicted values. This AI-driven approach offers a transformative alternative to traditional time-consuming and costly

Table 1: Encapsulation efficiency (EE%) of antioxidant-network components in liposomes—AI-predicted vs. experimentally measured values

| Antioxidant Component | AI Predicted EE (%) | Actual Measured EE (%) |
|---|---|---|
| Vitamin C | 26.46 - 54.40 | 48.51 ± 1.15 |
| Vitamin E | 86.16 - 88.50 | 86.16 ± 0.73 |
| Coenzyme Q10 | 77.58 - 96.00 | 82.03 ± 2.06 |
| Glutathione | 61.73 - 85.00 | 76.84 ± 0.81 |
| Alpha-Lipoic Acid | 54.00 - 95.58 | 61.68 ± 0.65 |

trial-and-error methods, significantly enhancing the efficiency of developing health functional food and pharmaceutical formulations.

Liposomes serve as an invaluable platform for encapsulating the diverse antioxidant network components, effectively addressing the challenge of integrating hydrophilic and lipophilic compounds into a single stable formulation. The unique bilayer structure of liposomes allows for the encapsulation of hydrophilic drugs in the aqueous core and hydrophobic drugs within the lipid bilayer. By leveraging AI to consider the distinct characteristics of each antioxidant, this research identified optimal co-encapsulation conditions, facilitating stable multi-antioxidant encapsulation within a nanocarrier. This approach helps overcome individual component instability and maximizes synergistic effects. Such multi-component co-encapsulation can improve the bioavailability of the antioxidant network, offering enhanced solutions for preventing and treating oxidative stress-related diseases.

A notable aspect of this study is the established iterative loop, where AI-proposed liposome formulations were experimentally validated, and the resulting feedback continuously refined the AI model's predictive accuracy. This human-AI collaboration significantly accelerates complex formulation development, maximizing AI's potential in formulation science. AI-based development not only reduces costs and time but also paves the way for personalized nutrition and therapies. Future applications could involve designing customized antioxidant liposome formulations based on individual biometric data, thereby accelerating personalized medicine in drug delivery.

Despite its strengths, this study has limitations. The dataset used for AI model training was somewhat limited in size and diversity, particularly regarding direct experimental data for all five co-encapsulated antioxidants. Expanding the dataset with more extensive and diverse multi-component co-encapsulation data could further improve the model's accuracy and generalization. Furthermore, the current model primarily focuses on predicting co-encapsulation efficiency. Future research should extend AI models to predict other critical quality attributes like stability, drug release kinetics, and in vivo efficacy, while also addressing practical challenges such as quality control, scale-up, and production costs. These efforts will solidify AI's role in revolutionizing multi-liposome drug formulation development for the antioxidant network.

## 5   Conclusion

This study demonstrates that AI-based prediction of encapsulation efficiency is a highly useful approach for accelerating the research and development of multi-component liposome formulations for the antioxidant network. Through AI models, we confirmed the possibility of effectively co-encapsulating five key components of the antioxidant network, including vitamin C, vitamin E, coenzyme Q10, glutathione, and alpha-lipoic acid, in a single liposome formulation. Specifically, liposomes actually prepared according to AI-recommended optimal compositions and manufacturing conditions exhibited excellent physicochemical properties and high co-encapsulation efficiency, experimentally validating the reliability of AI predictions. During this process, the predictive accuracy of the model was continuously improved through an iterative AI-human collaboration loop. Future research will include validation of liposome stability, release kinetics, and clinical applicability, thereby expanding the practical application scope of AI-based formulation development.

# Agents4Science AI Involvement Checklist

1. **Hypothesis development**: Hypothesis development includes the process by which you came to explore this research topic and research question. This can involve the background research performed by either researchers or by AI. This can also involve whether the idea was proposed by researchers or by AI.

   Answer: [C]

   Explanation: Liner AI synthesized literature on antioxidant networks, liposomes, and formulation informatics, and proposed a testable hypothesis: predicting and optimizing co-encapsulation efficiency (EE%) of a multi-antioxidant liposomal formulation from lipid composition and process parameters. The human researcher refined scope (materials allowed, process constraints, timelines), checked feasibility, and finalized endpoints and evaluation metrics.

2. **Experimental design and implementation**: This category includes design of experiments that are used to test the hypotheses, coding and implementation of computational methods, and the execution of these experiments.

   Answer: [B]

   Explanation: Liner AI suggested candidate lipid ratios, drug:lipid ranges, pH windows, and processing settings prioritized for higher EE%. The human researcher translated these into a practical lab protocol—e.g., thin-film hydration or ethanol-injection, followed by sonication/high-pressure homogenization—then executed liposome preparation and performed TEM, DLS, $\zeta$-potential, and EE% assays. Purification choices (dialysis/SEC) and QC were human-led; AI input was advisory.

3. **Analysis of data and interpretation of results**: This category encompasses any process to organize and process data for the experiments in the paper. It also includes interpretations of the results of the study.

   Answer: [C]

   Explanation: Liner AI handled preprocessing, model training (Random Forest, XGBoost, Neural Network), cross-validation, prediction, and feature-importance analysis (e.g., cholesterol fraction, hydrophilic/lipophilic class). The human verified assumptions, reconciled outliers with lab notes, and interpreted biological implications (differences between hydrophilic vs. lipophilic antioxidants). Experimental results were iteratively fed back to improve model performance.

4. **Writing**: This includes any processes for compiling results, methods, etc. into the final paper form. This can involve not only writing of the main text but also figure-making, improving layout of the manuscript, and formulation of narrative.

   Answer: [D]

   Explanation: Liner AI generated the outline, section text, methods descriptions, figure captions, and tables. The human researcher inserted measured EE% values, curated TEM/DLS figures, ensured methodological and regulatory compliance language, and edited for accuracy, clarity, and conference formatting.

5. **Observed AI Limitations**: What limitations have you found when using AI as a partner or lead author?

   Description: While Liner AI provided numerous literature-based examples and general formulation trends from prior studies, it did not generate directly actionable or experimentally validated liposome preparation conditions. The AI mainly summarized patterns from published research, leaving the translation into concrete, lab-ready protocols to the human researcher. This gap required substantial human expertise to bridge literature knowledge with practical experimental design.

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
